# Probing of the internal damage morphology in multilayered high-temperature superconducting wires

You-He Zhou [1,2], Cong Liu [1,2], Lei Shen[1,2] & Xingyi Zhang[1,2✉]

The second generation HTS wires have been used in many superconducting components of electrical engineering after they were fabricated. New challenge what we face to is how the damages occur in such wires with multi-layer structure under both mechanical and extreme environment, which also dominates their quality. In this work, a macroscale technique combined a real-time magneto-optical imaging with a cryogenic uniaxial-tensile loading system was established to investigate the damage behavior accompanied with magnetic flux evolution. Under a low speed of tensile strain, it was found that the local magnetic flux moves gradually to form intermittent multi-stack spindle penetrations, which corresponds to the cracks initiated from substrate and extend along both tape thickness and width directions, where the amorphous phases at the tip of cracks were also observed. The obtained results reveal the mechanism of damage formation and provide a potential orientation for improving mechanical quality of these wires.

[1] Key Laboratory of Mechanics on Disaster and Environment in Western China attached to the Ministry of Education of China, Lanzhou University, Lanzhou, Gansu, PR China. [2] Department of Mechanics and Engineering Sciences, College of Civil Engineering and Mechanics, Lanzhou University, Lanzhou, Gansu, PR China. ✉email: zhangxingyi@lzu.edu.cn

The application of high-temperature superconducting (HTS) materials, such as YBa$_2$Cu$_3$O$_{7-\delta}$ (YBCO) or Bi$_2$Sr$_2$Ca$_2$Cu$_3$O$_{10+\delta}$ (Bi-2223), has significantly lowered the cost of achieving zero resistance[1]. In comparison with Bi-based superconductors, YBCO superconductors have both a higher irreversible magnetic field and a larger critical current density; hence, they are potentially utilised for high-power applications[2–4]. However, the brittleness of this oxidation ceramic material makes it difficult to be directly fabricated. To overcome this deficiency, many revolutionary deposition techniques were developed to enable the YBCO layer to be deposited on either a biaxially textured (RABiTS) substrate or a substrate with an inclined-substrate-deposition (ISD), or buffer layer with iron-beam-assisted-deposition (IBAD), or through pulsed-laser deposition (PLD), or metal-organic chemical deposition (MOCVD), or reactive co-evaporation deposition (CSD) methods. After that, a protective coverage of metal layers is covered on the YBCO layer. Such preparation procedure makes the "Second Generation (2G)" HTS wires have the properties including flexibility and critical current significantly improved with long length (>1 km) and high performance (77 K, >1 MA/cm$^2$ in self-field)[5]. As we have known, the fabricated 2G HTS wires, named as coated conductors (CCs) with a multi-layered structure, have been worldwide produced by many companies, e.g., in the USA[6,7], Japan[8], and China[9]. Based on the CCs, the prototypes of transmission cables, motors, and magnets et al. have been manufactured, which provides a large market with great potential and wide prospects[10–12]. In these applications, there are two major forces that exert on the CCs: the thermal stress caused by the thermal mismatch of different material components and the Lorenz force generated from the interaction of high current and a strong magnetic field[13]. Because these forces are the main reason for deformation or even catastrophic destruction of superconducting components, the mechanical reliability of superconducting wires is essential to the operational and functional security of these devices[14]. To improve the mechanical properties of CCs, the mechanical behaviour under an applied strain should be well pre-known. However, the superconducting layer is thin and covered by other metal layers, which impedes its direct observation. At present, only some point-testing penetrating methods, such as X-ray and neutron diffraction techniques, can recognise lattice deformation. But such approaches all cannot work for the evolution of the strain-induced damage on a macro scale. The damages induced by a tensile strain are typically regarded as cracks that run through the thickness and propagate in a two-dimensional manner in the tape plane[15]. In addition, the extreme environments make it difficult for developing a device that can conduct a real-time observation of electromagnetic property variations during loading. The response recognition of such CCs damages upon external loading, especially the initiation and propagation morphology of the damage, has still been a prominent challenge, which serves as significant guidance for further finding an approach to improve their strain tolerance of the present engineering applications. Because there is no Faraday effect in superconducting materials, the magneto-optical imaging (MOI) technique is adopted by a magneto-optical (MO) indicator to observe the film on the surface of the superconducting sample, and the change in the magnetic flux is converted into the intensity variation of the MOI. With this approach, both the static and dynamic flux behaviours can be investigated[16,17]. Leiderer et al.[18,19] adopted this method to observe a velocity as high as 100 km/s during a flux avalanche. Feldmann et al.[20] revealed the influence of the grain angle on the critical current density in CCs. However, a few of such pioneering studies made preliminary attempts to acquire the response of magnetic flux during mechanical loading by the MOI method, such as studies on Bi-based superconducting wire with multi filaments[21] and YBCO micro bridges[22]. Till today, however, the dynamic flux behaviour of a macro-scale CC during loading has not been identified, making the evolution of strain-induced damage still unknown. In this study, a

controllable tension experiment on YBCO CCs with a macroscopic scale was established by using a cryogenic loading system combined with in situ and real-time MOI. The flux evolution behaviour varying with tensile strain was investigated, where some spot-like flux local motions occurred in the mixed-state region with a time scale of 30 ms at 40 K were observed when the strain reaches a critical value or over. After that, the increment of the strain causes the flux to break the critical state and to move into the Meissner region as sudden, multi-stack spindles. The velocities of these penetrations were obtained for the first time to our knowledge, and they ranged from 6 to 1059.3 µm/s, which suggests that the presented flux behaviour is distinctively different from that of the dendritic flux avalanche in superconductors. To characterise the damage morphology, the scanning electron microscope (SEM) imaging along the thickness direction of the YBCO layer was implemented by stepwise chemical etching of the superconducting layer. The obtained results reveal that the initial damages in the superconducting layer emerge as micro-cracks cleft by the substrate, then propagate along the directions of both width and thickness. Finally, the amorphous patterns were observed in the micro-crack tip using transmission electron microscopy (TEM), which can be considered as a mechanism for crack blunting and provide us a new way to effectively improve the mechanical properties of YBCO in future.

## Results

As displayed in Fig. 1a for HTS magnets, the HTS tape is wound into a pan-cake coil to supply a magnetic field along the axial direction. The exerted Lorentz force perpendicular to the tape surface induces hoop stress and therefore the tape is simultaneously subjected to tensile stress. The tape has a multilayer structure (Fig. 1b), which primarily consists of a superconducting layer (YBCO), a buffer layer, a substrate layer (Hastelloy-C276), and a cover layer (Ag). To probe the magnetic flux behaviour under tensile strain, a testing system was constructed, as shown in Fig. 1c, in which a MO microscope is cooperated in a cryogenic loading system. The sample and assembly details can be found in Supplementary materials Figs. S1 and S2.

A sequence of flux patterns with different tensile strains under zero-field-cooled (ZFC) conditions at 64 mT and 40 K is displayed in Fig. 2. In the sample without tension (Fig. 2a), a typical Meissner region is well formed by the shielding current, which indicates that this state-of-the-art tape has a well-aligned crystal grain texture. The asymmetric area in the mixed state is thought to be caused by the slightly non-uniform thickness of YBCO along the width. Several non-uniform flux penetrations on the left side are owing to the original damage induced by the cutting process. As the strain starts and increases, no obvious flux motion is observed until the strain reaches a critical value (Fig. 2b, c). It should be noted that when the strain is below the critical value, slight flux motions are observed, which may be caused by a slight change in the sample's physical properties. When the strain approaches to a critical value of 0.72% for this sample, the flux nucleates in the mixed-state region firstly, and some spot-like flux motions are subsequently generated (Fig. 2c). After a small strain is further increased, the pattern displays an intermittent spindle-like penetration in a small fraction of the specimen (Fig. 2d). When the strain is 0.78%, a large portion of the Meissner region is filled with magnetic flux in an intermittent manner (Fig. 2e), which leaves a small region that is not penetrated. Once the strain is 0.8%, the flux penetrations translate into sudden, global, multi-stack spindles, where the light (green) region sweeps away the Meissner phase (black region) in a large portion of this sample (Fig. 2f). This whole process can be attributed to the following main three stages: (i) when a moderate strain is applied, there is no prominent flux movement; (ii) when the strain reaches to a

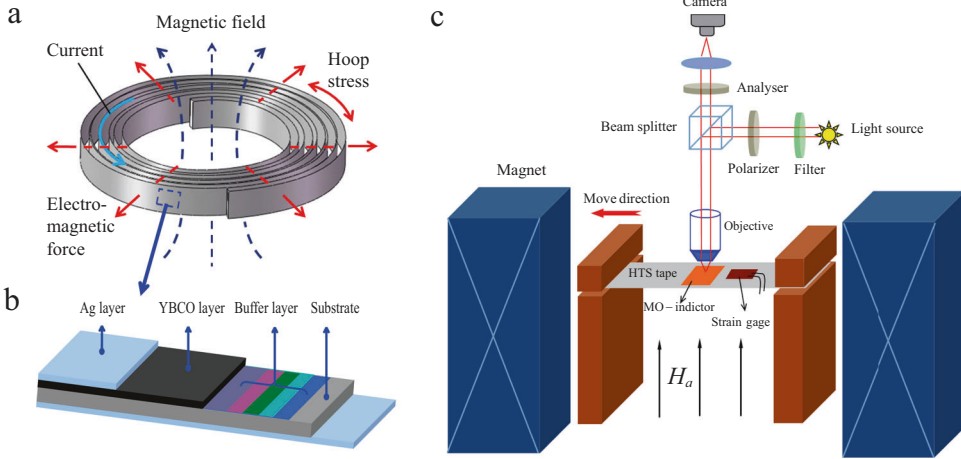

**Fig. 1 Schematic of high-temperature superconducting magnet, second generation high-temperature superconducting tape and testing device.**
**a** Schematic of a pan-cake coil constructed by the coated conductors, the electromagnetic force in the radial direction causes a hoop tensile stress along the tape length. **b** The sample has a multi-layer structure, consisting of a Hastelloy-C276 substrate of with a thickness of 50 μm, a buffer layer of 1 μm, $YBa_2Cu_3O_{7-\delta}$ (YBCO) layer of 1 μm, and Ag layer of 2 μm on both sides of the tape. **c** Schematic of the testing system, including a magneto-optical (MO) imaging microscope and cryogenic tensile loading device.

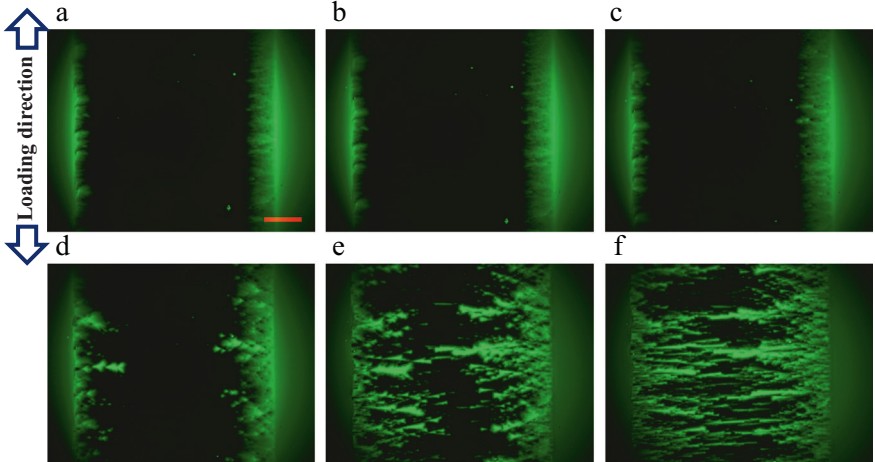

**Fig. 2 Flux patterns measured at different tensile strains.** The exerted strains under zero-field-cooled conditions at 64 mT and 40 K from (**a**) to (**f**) are 0%, 0.40%, 0.72%, 0.75%, 0.78%, and 0.80%, respectively. Here, the scale bar represents 500 μm.

critical value, the local spot-like flux motions firstly emerge in the mixed-state region; (iii) as the strain is over this critical value, more local flux motions occur and previously moved fluxes extend into the Meissner region. A similar phenomenon was also observed under ZFC conditions at 64 mT and 60 K and 77 K, respectively (see Supplementary materials Figs. S3 and S4).

In order to understand the electromagnetic property from starting to changing in the form of flux motion, here, the spot-like flux motion is firstly considered. Figure 3 shows the changing in the local flux motion with a strain from 0.70 to 0.72%. In this region, the total difference of the flux quantum motion is approximately formulated by $\Delta\Phi = 1.0 \times 10^6\ \Phi_0$, which implies that this spot-like flux motion can be considered as a type of collective vortex motion.

To obtain the time scale of the changing collective vortex motion, a high-speed camera was used to record the generating process. Dynamic nucleation and the collective vortex motion triggered by strain can be observed from a set of images of the local flux motion taken from 0–30 ms (Fig. 4a–d) and their grey level differences between each moment and the initiation (Fig. 4e–g). A similar phenomenon occurs with an increasing

time scale of 60 ms and 110 ms under ZFC conditions at 64 mT and 60 K and 77 K, respectively (see Supplementary materials Figs. S5 and S6). After spot-like flux movements are originally formed, a slight increment in the strain may further drive these fluxes to penetrate into the Meissner region. More compelling evidence for the speed distribution of the flux penetration was obtained from the time-resolved measurements. When the results are employed under the ZFC conditions at 64 mT and 40 K, for example, the maximum length of the flux penetration versus the loading time is illustrated in Fig. 5, from which one can see that several spot-like flux motions are activated at the initial moment. Then, the following process was recorded by the high-speed camera as the applied strain was increased from 0.70 to 0.78%. Here, the period of time in the analysis was used by 12.5 s, and the number of images was 2500, to which the penetration lengths are identified by the green circles and piecewise fitted by line with red rectangles (Fig. 5d). It is clearly found that the majority of the flux penetration is continuous along with the loading time, except for some jumping stages. It should be noted that these jumping phenomena are unique in the flux penetration triggered by an increasement of continuous strain loading. Except for these

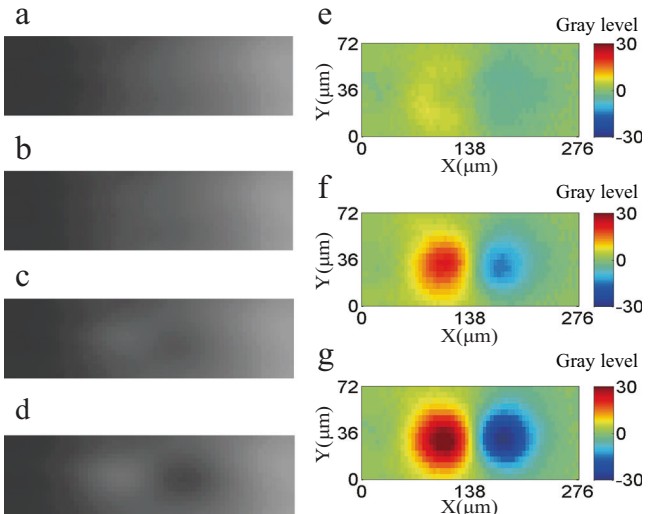

**Fig. 3 Local flux patterns and distribution of their magnetic field difference. a, b** Flux patterns with tensile strains of 0.70% and 0.72% under zero-field-cooled conditions at 64 mT and 40 K, respectively. The magnified images of the white boxes are inserted in the top left corner, and the scale bar represents 500 μm. **c** Their differential magnetic field distribution.

**Fig. 4 Local flux changing with loading time of the sample strained under zero-field-cooled conditions at 64 mT and 40 K. a–d** The magnifying images of the local area at 0, 20, 25, and 30 ms, respectively. **e–g** The grey level differences between each moment and the reference at the moment of 0 ms.

jumps, the average speeds for each continuous section are calculated, and their magnitude distribution is displayed in the inset of Fig. 5d, from where we find that the maximum velocity is 1059.3 μm/s, and the maximum penetration speed decreases with increasing temperature (see Supplementary materials Figs. S7 and S8).

To further investigate the penetration behaviour, the penetrated area was selected as a parameter for the statistical analysis. The differential areas in the neighbouring pair of images were calculated. The normalised occurrence probability of the penetrated area is shown in Fig. 6. Here, the period of time used in the statistics was 12.5 s, and the number of images was 2500. From Fig. 6, one sees that the first four data points were linearly fitted using the least-squares method with a slope of −1.43 ± 0.27. The similar statistical properties for the flux penetrations at 60 K and 77 K, respectively, with ZFC conditions at 64 mT are also gained, and they are provided in the Supplementary materials (see Supplementary materials Figs. S9 and S10).

To determine where the damage occurred in corresponding the flux motion process, both chemical etching[23] and SEM imaging methods were selected. When the sample was firstly strained to 0.73% under ZFC conditions at 64 mT and 40 K, the part of the corresponding MO image is displayed in Fig. 7a. Then the sample was removed out from the above test environment, and the Ag layer was chemically removed by a mixture of hydrogen peroxide and ammonia monohydrate with a volume

ratio of 1:4. After that, several cracks were evidently observed at the surface of the YBCO layer, where distinct flux penetrations occurred. That is, no cracks on the surface of the YBCO layer can be directly observed in the strained sample, but the MO image just has several corresponding spot-like flux motions as the case shown in Fig. 2c.

To compare the response of the magnetic field in a sample with a crack throughout the YBCO layer, an artificial crack was generated by an indentation with a thin blade on the tape surface (close to the YBCO layer). Then, the flux penetration lengths under the same experimental conditions for the 0.76% strained sample and the sample with the artificial crack are compared in Fig. 8.

From Fig. 8, it is evidently found that the flux in the strained sample enters into the Meissner region more sensibly than that with an artificial crack, which suggests the fact that the damage is not a type of crack extended throughout the thickness of the YBCO layer because the flux front at the strain-induced damage strongly depends on the applied magnetic field.

To further determine the damage model, a 0.75% strained sample was firstly cut into pieces with dimensions of 6 × 3 mm, and the Ag layer was then chemically removed as done previously. Before further etching the YBCO layer, a scanning electron microscopy (SEM) image was obtained. After that, the etching was conducted again, and the piece was placed in a 2% Br/ethanol solution for 5 s to partly etch the YBCO layer[24] for subsequent SEM imaging, to which this procedure was repeated stepwise. Finally, a set of surface SEM images of etched YBCO layers with different thicknesses were gained. Based on such images, the areas where the flux penetrations occurred were carefully examined. From them, it is surprised for us that in some areas, several cracks can be clearly observed just after the removal of the Ag layer. However, there is no damage in some areas. With the reduction of the YBCO thickness by etching, the SEM imaging plane in the YBCO layer becomes close to the buffer layer. In such case, it is found that some cracks emerge and their lengths increase, as shown in Fig. 9a and in the Supplementary materials Fig. S16. By means of the closer observation of these emergent crack tips, moreover, the intermittent connections are displayed, which are depicted in Fig. 9b. Then, a small part of this connection is cut out by the focused iron beam (FIB) approach, and TEM is used to observe a crack tip in the cut piece, from which the observed images are displayed in Fig. 9c. Here, we clearly observe the fact that the crack is a cleavage type, and it initiates from the substrate layer, then crosses the buffer layer, and finally terminates in the YBCO layer, as shown in Fig. 9d. In Fig. 9e, two small areas are focused on the region enclosed the crack tip: one labelled "1" is near the crack tip (Fig. 9f), and the other labelled "2" is at the boundary of the crack tip (Fig. 9g). From them, we find that only a crystalline phase is appeared in the former case, while both amorphous and crystalline phases in the latter one are separated by the boundary. The corresponding elementary analysis results are given in Supplementary materials Fig. S17.

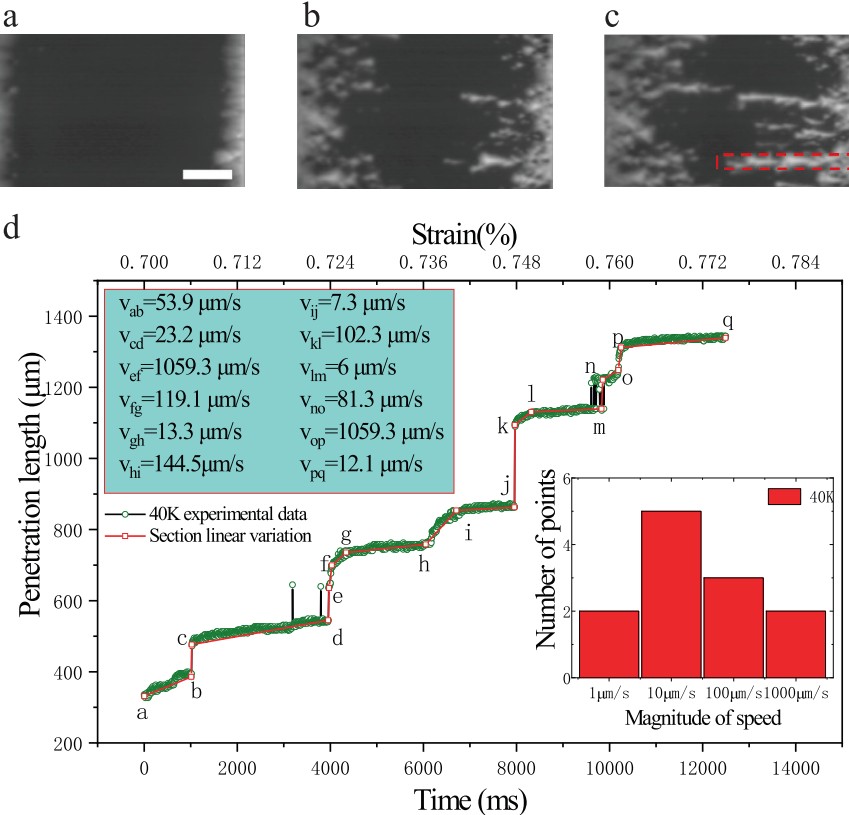

**Fig. 5 The characteristic curve of flux penetration length versus loading time. a–c** Flux evolutions at 0, 8150, and 12,500 ms, respectively. Here, the scale bar is 500 μm. **d** Penetration length in the region marked by a red rectangle in (**c**) depends on the loading time. Some jumping stages are observed, and the average speeds for each continuous section from the red curve are calculated. The inserted is their magnitude distribution.

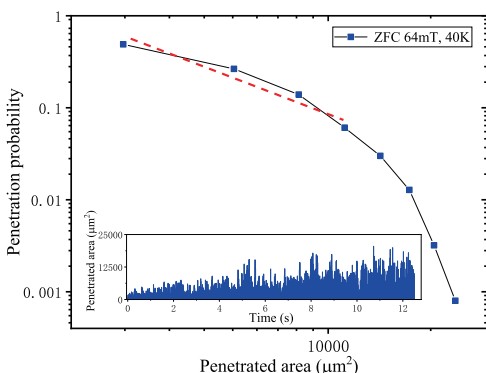

**Fig. 6 Penetration probability of flux penetrated area.** The curve corresponds to the statistics of the inserted penetration area with a time interval of 5 ms, where a straight line with a slope of −1.43 ± 0.27 is added as a reference for visualisation.

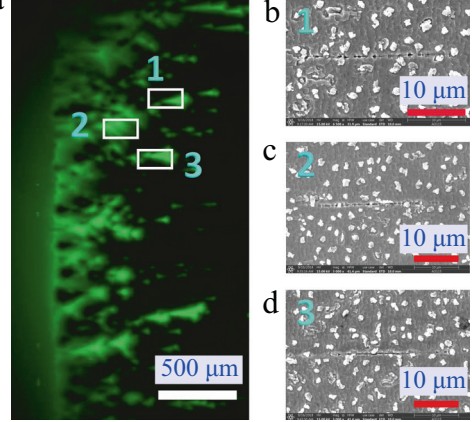

**Fig. 7 MO image and SEM images at the local microlevel. a** The MO image of a 0.73% strained sample. **b–d** Three microlevel local SEM images from (**a**), where their damages were observed in these flux penetrated areas.

## Discussion

We conducted a probing experiment to visualise the internal damage induced by a tunable strain method in the serving environment for the superconducting CCs used in an application scale. By means of this approach, the challenge for recognising the interior damage morphology under an applied tensile strain has been resolved. By comparing point-testing methods, such as X-ray and neutron diffraction techniques, our proposed method can provide information on damage evolution at a macro scale. In this extent, there is no difficulty for us to use the presented experimental approach for conducting the necessary tests of the damages in the 2G HTS wires and to further provide guidance for

the enhancement of the mechanical properties of 2G HTS wires. In addition, the constructed measurement system can be also employed to investigate the influence of strain on the critical current density under either a self-field or externally applied magnetic field, which is very important to fully understand the mechanism of the strain effect[25,26].

The MOI results show that when the applied strain reaches a certain value, the spot-shaped local flux motion appears, where the formation time is in the order of tens of ms, and the magnitude of the magnetic field changing is in the order of $10^6$ magnetic flux

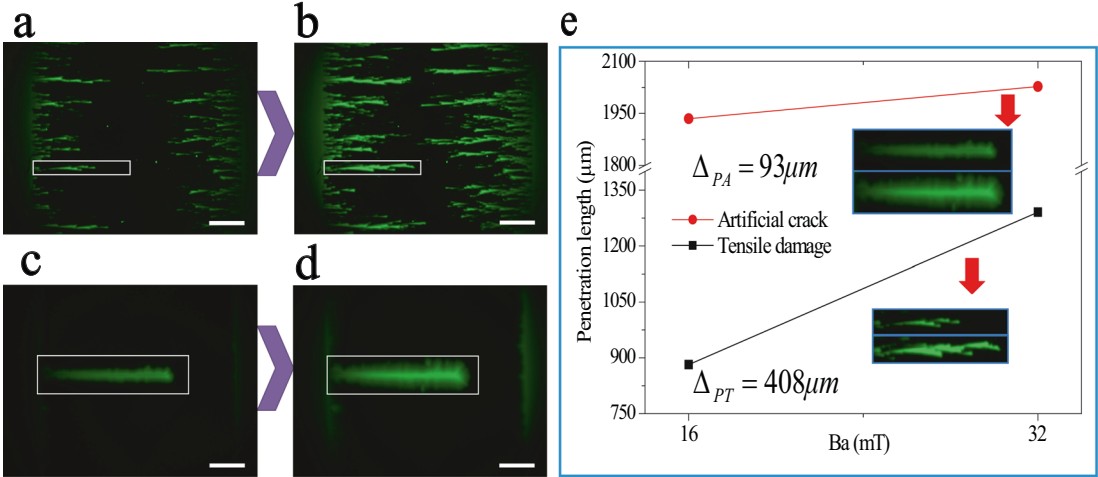

**Fig. 8 A comparison of magnetic field responses between samples with tensile damage and an artificial crack.** Here, the scale bar represents 500 μm. **a**, **b** Flux images of the strained sample (0.76%) under zero-field-cooled (ZFC) conditions 16 mT and 32 mT at 40 K, respectively. **c**, **d** Represent flux images of the sample with an artificial crack of length 1746 μm under ZFC conditions 16 mT and 32 mT at 40 K, respectively. **e** The max penetration length varies with the applied field in both samples. The inset images are the regions of interest in our study.

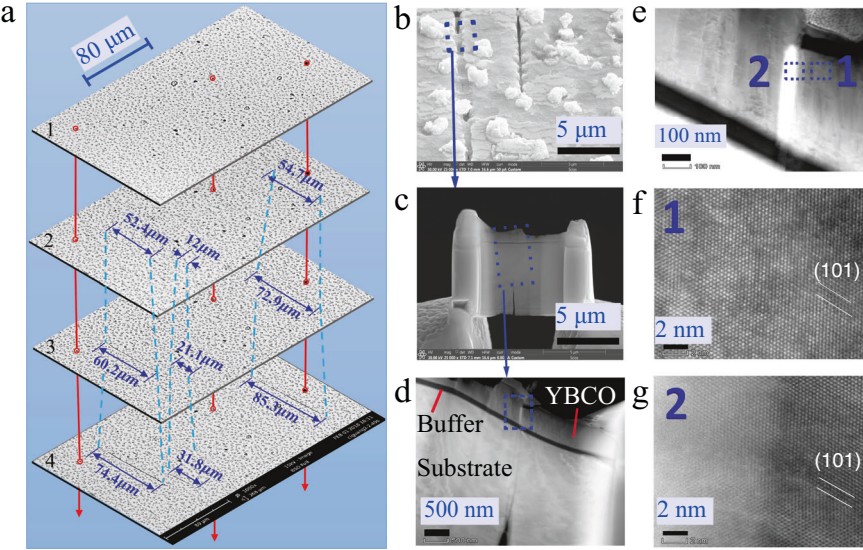

**Fig. 9 Scanning electron microscopy (SEM) images of the etched sample after a strain of 0.75%. a** A set of SEM images from 1–4 of etching times of 0, 10, 15, and 40 s. The total time for the removal of YBCO is about 270 s, and the speed of the etching process is 3.7 nm/s. **b** The SEM images of crack (etched for 40 s). **c** Small cut part containing local crack is performed by the focused iron beam method. **d** The magnified image of rectangle box contained in (**c**). **e** The magnified local crack tip prepared for transmission electron microscopy (TEM) imaging. **f** The TEM image of crystalline phase in the area far from crack. **g** The TEM image visually contains both the amorphous and crystalline phases in and near the crack tip, respectively.

quanta. As the temperature increased, the formation time gradually increases too. In general, large flux penetration into the Meissner region occurs in the form of a flux avalanche caused by magnetic and thermal instability. The flux avalanche has three characteristics: (i) the randomness of the position where the magnetic flux collapses, i.e., the same experimental conditions result in different magnetic flux diffusions; (ii) the speed of magnetic flux diffusion is extremely fast, or the speed of magnetic flux diffusion in YBCO films can reach to 180 km/s;[19] (iii) the magnetic flux penetration is displayed by a discontinuous manner. Corresponding to it, both the dendritic avalanche and the intermittent of one-dimensional penetration exhibit self-organised critical (SOC) behavior[27,28], and the power index of its characteristic statistics ranges from −1.8 to −1.3. In our study, the power indexes in the statistical results of the penetration area for 40 K and 60 K are in the region of (−1.8, −1.3), except for the case

at 77 K. At the same time, our observation of the magnetic flux diffusion also has the main characteristics of SOC. However, the low–speed penetration feature indicates a mechanism being significantly different from the magnetic and thermal instability.

From a structural perspective, the magnetic flux is relatively stable before the strain is approached to a critical value, which tells us that the relative small strain does not significantly affect its electromagnetic properties. This is because the superconducting layer deformed elastically such that no damage occurs, which was confirmed by the X-ray diffraction[29] and neutron diffraction results[30]. When the strain is approached to and over the critical value, our SEM and TEM results observe that the elastic deformation is mediated by generating a cleavage along the thickness direction, which can lower the pinning potential and movements of a local flux as the collective de-pinning vortices occur. Further, a larger strain may release the elastic energy by extending and

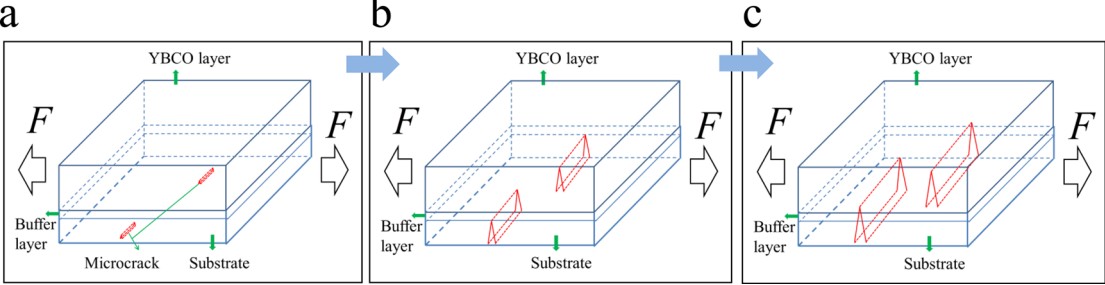

**Fig. 10 Schematic of damage formation process during tensile strain. a** Micro cracks in a cleavage-type started from the substrate. **b** Micro cracks propagate into the buffer layer with an increasing strain. **c** Microcracks penetrate into the superconducting layer along the directions of both the thickness and width when the strain further increases.

propagating the cracks along both the thickness and width directions of the CC, as shown in Fig. 9a, in which the Meissner region is broken and a large amount of flux penetrates. The magnified results at the crack tip along the thickness direction clearly demonstrate that the crack is blunted by the amorphous phase, which is first found in CCs, apart from biomaterials[31]. Hence, a portion of the YBCO layer thickness remains undamaged. Therefore, it can still carry a superconducting current, thereby shielding the flux in a low magnetic field, but not in a high magnetic field, as indicated in Fig. 8 and Supplementary materials S3. Such experimental results can also qualitatively explain the magnetic-field dependence of the degradation of the critical current with a large strain[32]. Based on these analyses, a schematic of the damage formation process is shown in Fig. 10.

The existing studies on ISD wires technique have suggested that discontinuous yielding dominates the irreversible strain limit of the wires, and the Luders band of the annealed Hastelloy C-276 substrate was formed near the clamps by shearing yielding on a macro scale[33]. Here, the Luders band was not observed, and no cracks were observed on the Ag surface of the strained tape, even strain up to 1%. It should be noted that the flux penetration behaviour in this study is very different from the result obtained by the short micro YBCO bridge[22]. The main reason is that the MOG location in this study is 25 mm away from the two clamps. Hence, the effect of the two clamps on the tensile stress distribution along the tape width at the MO imaging area is minimised, which is very important for providing a reasonable damage analysis during pure tension. As a result, the Luders bands on the substrate in the tension condition were not observed in the MO imaging area, as shown in Fig. S18. Moreover, the idea of characterising internal damages based on magnetic characteristics is also useful for other magnetic functional materials, such as multi-layered magnetic films, where the damage of the magnetic layer caused by strain can be experimentally analysed by the response of the magnetic field.

The strain-triggered flux evolution combined with a structural damage analysis reveals that the damage starts from the substrate in 2G HTS wires, and a new mechanism for crack blunting in the amorphous phase at the tip of the crack in the YBCO material is found. However, the reason for the formation of the amorphous phase is still unknown. Whether it is caused by the deposition method, or the multi-layer structure, or the property of the YBCO material itself remains an open question. Because those amorphous phases observed in biomaterials play a key role in enhancing toughness[29], research on the toughness enhancement of amorphous phases in superconducting materials is required in the future. In summary, with the strong need to improve the strain tolerance of superconducting brittle materials in HTS wires for large-scale applications, this study provides a new tool to insight the strain influence on 2G HTS wires.

## Methods

The SF12050 tape (Super-Power Inc., USA) has a multi-layered structure. The buffer layer was deposited on the substrate using the IBAD approach, and the YBCO layer was deposited on top of the buffer layer via the MOCVD method. Both surfaces of the substrate and YBCO layer were then covered with a 2 μm Ag layer. The sample details are presented in Fig. S1. These tapes were mechanically cut into a 3 mm width and 100 mm length in the experiment for full-width MOG.

To investigate the flux behaviour with tension in a real-time full-field environment, a new system combining MOG and cryogenic multi-field loading techniques was constructed. In such case, the 2 G HTS tape can be stretched in an applied magnetic field, and the real-time behaviour of the flux can be mapped through the MOI system, while the strain can be controlled and measured simultaneously from the strain gauge glued on the surface of the tape. The sample assembly is shown in Fig. S2. During the experiment, the chamber was firstly vacuumed by a vortex pump and molecular pump at ~$10^{-3}$ Pa, and the sample was then cooled through the clamps. Next, a magnetic field was applied along the direction normal to the tape plane and the tensile loading was subsequently exerted, while the strain was recorded by the strain gauge (KFL-1-120-C1-11). A commercial polarised microscope (Olympus BXFM) was combined with a built-in objective (LMPLFLAN5×). A typical MO indictor with an MO layer thickness of 5 μm associated with a Faraday coefficient of 0.012°μm$^{-1}$ mT$^{-1}$, a 0.5-mm thick substrate, and an Al mirror of about 1μm is used for flux imaging. The recording camera for the static strain state is MC-50N made in China, and the dynamic strain state is NR4-S1 made in the USA.

## Data availability

Source data are provided with this paper. https://doi.org/10.6084/m9.figshare.14361206.v1

## Code availability

The code used to perform the image process is available upon reasonable request to the corresponding author.

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

## Acknowledgements

This work is supported by the Fund of Natural Science Foundation of China (No. 11902130, 11872196), and by the 111 Project (B14044). Prof. Hongtao Wang at the Zhejiang University of China, Prof. Anmin Nie at the Yanshan University of China, and Dr. Donghua Yue at China Academy of Engineering Physics gave their helps for the TEM experiments in this research. The authors sincerely and deeply acknowledge these supports and helps. We also acknowledge the helpful discussions with Prof. Xiaoyan Li at the Tsinghua University of China.

## Author contributions

Y.H.Z. and X.Y.Z. conceived and supervised the study. X.Y.Z. and C.L. designed the experimental device. C.L. performed experiments and analysed the data with the help of X.Y.Z. L.S. performed some SEM experiments. C.L. and X.Y.Z. wrote the paper. Y.H.Z. examined the experimental data processing, and conducted the final revision of the paper.

## Competing interests

The authors declare no competing interests.
