## [Peer Review File · Nature Communications]

Reviewers' comments:

Reviewer #2 (Remarks to the Author):

In all of high temperature superconducting materials, REBCO coated conductors-CCs, as the 2nd generation conductors, have been the mostly attractive conductor mainly because their higher critical current density and no sharply declining (something like the case of low temperature conductors, typically Nb-based alloys) with applied magnetic field. Even though, to push it to many potential applications, there are some challenges such as brittle, grain-boundary- angle dependent, anisotropic and crack properties. Especially for crack behaviors, there are rarely studied because of the two challenge, one is that superconducting layer is covered the by the metal layer impeding the direct observation and locating the position of the damage, the second is the common techniques are hard and limited for realizing visualization the morphology of the damage like cracks in extreme experimental conditions of low temperature, vacuum and magnetic environment.

Hence in my view, the main contribution of this article is that the authors have constructed and developed a novel magneto-optical combined cryogenic loading system in extreme environment, which is of significant for studying the electro-mechanical properties under tensile loading strain in CCs with width close to engineering scale, only through which the coupled behavior between the flux evolution and strain can be revealed in a real-time, space-resolving way. According to the proposed method of optical probing internal damage, series of new phenomena for flux motion have been obtained including collective vortex motion, intermittent flux propagations with low speeds, which expand the understanding of the magnetic flux dynamics of HTS materials. Notably, a new mechanism for crack blunting by amorphous phase at the tip of the crack in superconducting material is firstly found, indicating a new perspective for enhancing mechanical properties of brittle ceramic materials in layer, flexible CCs.

I think the novelty of finding in this article reaches the requirement of Nature Communication. This work significantly enhances the understanding of electro-mechanical behavior, and also be directly useful to improve the mechanical 2 performance of CCs, and thus, it could attract more people who involve in superconducting materials and/or related industrial fields.

Nevertheless, there are some concerned questions or comments need to be answered and solved before going into publication. They are listed as follows:

Q1: In MO image results, the center part are clear but color turn into darken at the corner, but the width of mixed state seems the same along the length of the CC, please give the explanation.

Q2: I notice that the widths of mix state along the edges in all CC samples are not the same, what is the cause?

Q3: the directions of tension and width should be marked in MO results for a better visualization.

Q4: In the tensile test, as we know, uniform and stability of the temperature are important. Is there a temperature difference between the CC sample and the chuck? Suppose that there is such a temperature difference, whether there is an effect on the experimental results? The authors need to provide more details of temperature information.

Q5: In other previous studies, the irreversible strain in the sample without Cu stabilizer is about 0.4%-0.45% below 0.7%. Hence the crack is expected, in the MO result, in the strain range between 0.4%-0.45%, no obvious flux difference are observed. If there is a relation between the irreversible strain and the critical strain that trigger the flux local movement. A detailed discussion about this should be provided.

Q6: The damage induced by tensile strain should be randomly distributed along the tape but in the MO result, local flux motion always started in the mixed region. i.e. from the edge of the sample, this is conflicting to the common understanding, please explain this phenomenon.

Q7: The amorphous phase in the tip of YBCO is a novel found, how much that 3 donates the toughness of YBCO, if can be done please added, if not, please add more references and give some indications of possible experimental completion for further study.

Q8: I think this method is effective for structure broken analysis of other function films especially magnetic films in multiple physical fields. The authors should add some expansion of the present method in the discussion section.

Q9: In page 2, author lost mention of technique of rolling assisted biaxially textured substrate (RABITS).

Q10: In Supplementary Materials, I noticed the strain gage is behind a piece of copperplate, please explain the purpose.

Q11: In the discussion section, the power index of its characteristic statistics meets from 1.3 to 1.8 should be from -1.8 to -1.3, author should check that.

Q12: I find the slope of time statistics of penetration area is decreased with elevated temperature, please give the reason.

Q13: In figure S12, Author finds no Luders bands existed in the Hastelloy substrate used, please attach the possible explanation behind.

Q14: Some sentence need to be modified for langue fluency. Such as in the abstract "Not until a novel approach has developed by depositing YBCO layers on either a flexible metal or alloy substrate and then following coverage of protective layers to realize a long-length fabrication, have the mechanical properties of such "Second Generation (2G)" HTS wires significantly improved."

Reviewer #3 (Remarks to the Author):

This manuscript reported a device which combines magneto-optical imaging (MOI) and cryogenic multi-field loading techniques to probe the real-time sample damage of "2G" HTS wires in the runtime environment. The detection of real-time damage morphology of the YBCO layer (2G HTS) under tension strain is a challenging issue and it is crucial to their applications. The reported device can observe the vortex motion under runtime environments and in a real-time way. With this device, the authors first observed the point-like flux nucleation and spindle shape flux penetration induced by the stretching strain in the 2G HTS wires. The behaviors are indeed different from the flux avalanche triggered by traditional magnetic and thermal instability, which help us to understand the vortex dynamics of type-II superconductors.

The authors claimed that their device can also probe the sample damage evolution of "2G" HTS wires in the runtime environment. However, the sample damage is actually measured by SEM or TEM when

the sample is removed from the device and chemically etched. Thus, such an observation of sample damage is neither "real-time", nor optical probing. The title of the manuscript is quite misleading to the readers and it should be modified.

Such a MOI method to observe vortex motion (even vortex avalanches) in runtime environments has already been reported previously and it is widely used in this field (for example, see Refs. 16-20). The MOI techniques combining a mechanical loading device in the cryogenic environment has also been reported previously (see Ref.21,22). I do not think the device proposed in this manuscript has any significant advances in probing the vortex motion or sample damage.

The damage model (shown in Fig.10) is just a conjecture. It requires more direct experimental evidence to support it.

By the way, there are too many language problems or typos in the manuscript. Here I only list some of them.

Line 74, "the reveal the evolution...";

Line 80, "The responding of CCs", I cannot find the definition of "CCs".

Line 90, "100 Km/s" should be "100 km/s".

Line 153, "it reaches at a critical value"

Line 167, "can be sort into".

Line 212, "would further drove the flux"

Line 226, "speeds of etch continuous section" ??

Line 286, "can clearly see".

Line 442, "form strain gage".

...

There are also some misleading expressions in the manuscript.

Responses to referees

We are very grateful to the reviewers for taking the time and effort to review our paper. Their positive comments and insightful suggestions have helped us to improve the manuscript. According to their comments and suggestions, we conducted additional experiments and analysis to interpret the present experimental data. The point-by-point responses to each critique are provided below.

To reviewer 2

Query 1: In MO image results, the center part are clear but color turn into darken at the corner, but the width of mixed-state seems the same along the length of the CC, please give the explanation.

Answer: To obtain a microscopic MO image in a vacuum chamber, a focusing module is designed, which contains a built-in objective. As a result, the non-uniform illumination makes the MO image darken at the corner.

Query 2: I notice that the widths of mixed-state along the edges in all CC samples are not the same, what is the cause?

Answer: It is due to the little non-uniform thickness of YBCO layer in the manufacturing process. The sample used is cut into 4mm width from an intact sample of width of 12 mm, so in some samples, one edge is its original, and the other is a cutting edge, resulting in a little difference

between two widths of regions in mixed-state along the edges.

Query 3: the directions of tension and width should be marked in MO results for a better visualization.

Answer: The direction of tension has been marked in all MO results.

Query 4: In the tensile test, as we know, uniform and stability of the temperature are important. Is there a temperature difference between the CC sample and the chuck? Suppose that there is such a temperature difference, whether there is an effect on the experimental results? The authors need to provide more details of temperature information.

Answer: The temperature is uniform along the tape length direction after the thermal balance, as shown in figure 1. But there is a temperature difference between the sample and the clamps due to the thermal contact resistance, as shown in figure 2. If the temperature is not uniform along the sample length, the critical current density would not be a constant along the tape length, resulting in a different width of mixed-state region along the tape length.

Figure 1. The temperatures at different positions on the sample during cooling process. (a) Position “1” and “2” denote the temperatures at 1 cm away from the two clamps, respectively. (b) Their temperature difference depends on cooling time.

Figure 2. Temperature variation during controlling process.

Query 5: In other previous studies, the irreversible strain in the sample without Cu stabilizer is about 0.4%-0.45% below 0.7%. Hence the crack is expected, in the MO result, in the strain range between 0.4%-0.45%, no obvious flux difference are observed. If there is a relation between the irreversible strain and the critical strain that trigger the flux local movement. A detailed discussion about this should be provided.

Answer: In the critical current test, the current density distribution is almost uniform along the tape width. Hence any little micro damage

forms in the YBCO layer along the tape length would reduce the critical current. In our experiment, we only implement MO imaging on part of the entire sample, so there may be a difference between the irreversible strain and strain that triggers the local movement. However this is just an inference, a detail experiment would be designed and conducted in our next study on the relationship between these two featured strains.

Query 6: The damage induced by tensile strain should be randomly distributed along the tape but in the MO result, local flux motion always started in the mixed-state region. i.e. from the edge of the sample, this is conflicting to the common understanding, please explain this phenomenon.

Answer: The reason is that the induced current by applied magnetic field in our experiment is not uniform along the tape width, a typical currents distribution can be found in the reference (E. H. Brandt et al. ,Europhys. Lett. 22 , 735 (1993)), where current density in the mixed-state region researches its critical value, but is lower than the critical value in the Meissner region. We have found the damage is not throughout the thickness of the YBCO layer at strain level that local flux motion takes place, so in the Meissner region, hence there is still some portion of thickness remains undamaged so that it can still carry superconducting current that shielding the flux out in low magnetic field. As a result no

flux penetrations are simultaneously observed in the Meissner region.

Query 7: The amorphous phase in the tip of YBCO is a novel found, how much that donates the toughness of YBCO, if can be done please added, if not, please add more references and give some indications of possible experimental completion for further study.

Answer: The contribution of amorphous phase to the YBCO toughness is not measured to now, and it is expected to be quantitatively measured by micro-TEM loading experiments.

Query 8: I think this method is effective for structure broken analysis of other function films especially magnetic films in multiple physical fields. The authors should add some expansion of the present method in the discussion section.

Answer: “The idea of characterizing internal damages based on magnetic characteristics is also useful for other magnetic function materials, such as the multilayered magnetic films, of which the damage of magnetic layer caused by strain can be analyzed through the response of magnetic field.” has been added in the part of discussion.

Query 9: In page 2, author lost mention of technique of rolling assisted biaxially textured substrate (RABiTS).

Answer: “rolling assisted biaxially textured substrate (RABiTS)” has been added in the text as “Not until revolutionary deposition techniques are developed, in which YBCO layer is deposited on either a biaxially textured (RABiTS) substrate or substrate with an inclined-substrate-deposition (ISD), or iron-beam-assisted-deposition (IBAD) buffer layer,…”

Query 10: In Supplementary Materials, I noticed the strain gage is behind a piece of copperplate, please explain the purpose.

Answer: The function of the copperplate is used as a thermal radiation shield to keep the temperature at position where the strain gage is glued constant.

Query 11: In the discussion section, the power index of its characteristic statistics meets from 1.3 to 1.8 should be from -1.8 to -1.3, author should check that.

Answer: It has been corrected as “from -1.8 to -1.3”.

Query 12: I find the slope of time statistics of penetration area is decreased with elevated temperature, please give the reason.

Answer: With temperature elevating, the critical current density reduces, as a result, the driving force on the flux vortex is reduced so the

possibility of fast large magnetic flux movements is lowered, and the slope of time statistics of penetration area is decreased.

Query 13: In figure S12, Author finds no Luders bands existed in the Hastelloy substrate used, please attach the possible explanation behind.

Answer: Luders bands were only found near the clamps for long length sample. In our experiment device of which the inner space is as large as 150mm, the gauge length and width of sample are 80 mm and 3mm, respectively. The width is closed to the engineering application scale. The magneto-optical imaging place is 25 mm far from the two clamps, and hence the effect of two clamps on tensile stress distribution at MO imaging area is minimized, which is important to give a reasonable damage analysis during pure tension. As a result, Luders bands on substrate in tension are not observed in the MO imaging area.

Query 14: Some sentences need to be modified for language fluency. Such as in the abstract “Not until a novel approach has developed by depositing YBCO layers on either a flexible metal or alloy substrate and then following coverage of protective layers to realize a long-length fabrication, have the mechanical properties of such “Second Generation (2G)” HTS wires significantly improved.”

Answer: Sentence “Not until a novel approach has developed by

depositing YBCO layers on either a flexible metal or alloy substrate and then following coverage of protective layers to realize a long-length fabrication, have the mechanical properties of such “Second Generation (2G)” HTS wires significantly improved.” has been corrected as “Not until various novel YBCO depositing approaches on either a flexible metal or alloy substrate have been developed for a long-length fabrication, have the mechanical properties of such “Second Generation (2G)” HTS wires significantly improved.” Other sentences have also been checked and modified for language fluency.

To reviewer 3

Query1 : There are too many language problems or typos in the manuscript. Here I only list some of them.

Line 74, "the reveal the evolution...";

Line 80, "The responding of CCs", I cannot find the definition of "CCs".

Line 90, "100 Km/s" should be "100 km/s".

Line 153, "it reaches at a critical value"

Line 167, "can be sort into".

Line 212, "would further drove the flux"

Line 226, "speeds of etch continuous section" ??

Line 286, "can clearly see".

Line 442, "form strain gage"....

There are also some misleading expressions in the manuscript.

Answer:

1. "the reveal the evolution..." has been corrected as "present the evolution of..."
2. The definition of "CCs" are added in the text "Since then, 2G HTS wires, known as coated conductors (CCs) with a multi-layered structure, are worldwide produced by many companies from USA, Japan, and China et. al."
3. "100 Km/s" has been corrected as "100 km/s".
4. "it reaches at a critical value" has been corrected as "it reaches a critical value".
5. "can be sort into" has been corrected as "can be sorted into".
6. "would further drove the flux" has been corrected as "would further drive the flux"
7. "speeds of etch continuous section" has been corrected as "speeds of each continuous section"
8. "some damages can clearly see" has been corrected as "some damages can be clearly seen".
9. "form strain gage" has been corrected as "from strain gage".
10. Other misleading expressions in the manuscript have also been checked and corrected.

Query2 : The title of the manuscript is quite misleading to the readers and it should be modified.

Answer: The title has been corrected as “Probing of the internal damage morphology in multilayered high-temperature superconducting wires”.

Query3 : Such a MOI method to observe vortex motion (even vortex avalanches) in runtime environments has already been reported previously and it is widely used in this field (for example, see Refs. 16-20). The MOI techniques combining a mechanical loading device in the cryogenic environment has also been reported previously (see Ref.21,22). I do not think the device proposed in this manuscript has any significant advances in probing the vortex motion or sample damage.

Answer :

1. In works of references from 16 to 20, none of the flux/vortex motions are driven by the applied mechanical strain. The mechanism of flux motion behind in our work is totally different from those works. The intact magnetic flux penetrating process under tensile strain are firstly provided and reported.

2. In reference 21, the dynamic flux behavior of a macro-scale (engineering-scale with several “mm”) 2G HTS wire during loading in zero-field cooled magnetic field cannot be conducted by their

experimental devices because of the limits of space of device, they must remove the magnet when applied loading force, so only field-cooled experiments can be implemented. In reference 22, they provided the results of micro-bridge (150 μm width), and attributed the flux penetrating behavior to the Luders bands. We think the magnitude of the width of 2G HTS coated conductors in practice is in the order of “mm” not “ μm ”, the result of micro bridge cannot be directly extrapolated to samples with width of order of “mm”. Besides, the stress distribution along the width direction is not uniform due to the short length of the sample (20 mm long) used in their tensile test, in that case, the influence of two clamps on stress distribution along the width direction cannot be ignored, as a result, the Luders bands form, which has been also verified in the reference 35, author just found the Luders bands were only found near the clamps for long length sample. In our experiment the inner space of device is as large as 150mm, the gauge length and width of sample are 80 mm and 3mm, respectively. The width is closed to the engineering application scale. The magneto-optical imaging place is 25 mm far from the two clamps, and hence the effect of two clamps on tensile stress distribution at MO imaging area is minimized, which is important to give a reasonable damage analysis during pure tension. As a result, Luders bands on substrate in tension are not observed in the MO imaging area, as shown in figure S12. Most of all, in their works, damages are thought to

be thickness independent, therefore the thickness dependent of crack and amorphous phase in the crack tip are not found and reported, which is one of the novelty in our work.

Query 4: “The damage model (shown in Fig.10) is just a conjecture. It requires more direct experimental evidence to support it”.

Answer: Additional experiments have been added as in supplementary materials of parts S3 and S4 to support the damage model.

REVIEWER COMMENTS

Reviewer #2 (Remarks to the Author):

Review comments for the manuscript NCOMMS-20-10704A-Z

On the basis of reviewers' comments, the authors have made sufficient modifications to the manuscript and answered the questions that reviewers concerned about. And some new additional experiments are very necessary for further supporting the conclusions of this work. As a whole, this revised version meets the publishing requirements of Nature Communications. But before publication, there are still one question and some detailed problems for this revised version, and besides authors should refine the English writing further.

1. For the first query of the reviewer, "In MO image results, the center part is clear but color turn into darkening at the corner, but the width of mixed-state seems the same along the length of the CC, please give the explanation." The explanation given by the authors is that it is caused by the non-uniform illumination, I approve this response, but I still want to know how to eliminate it? Figure 3 displayed the results of local magnetic field variation. Do the authors consider eliminating the influences of non-uniform illumination on the present results? Please clear it.
2. In P18, the discussion of the relationship between "spot-like flux motion" and the "collective flux motion" should be deleted. Since the present flux motion caused by strain is a new experimental phenomenon, the reviewer considers that the experimental observations of the spot-like flux motion in this paper are significantly different from the classical flux creep considered in the condensed matter physics. If the authors want to retain this discussion, please explain in depth.
3. There are still some syntax and semantics in this revised version, In P5, application should be applications, move should be moves, etc. Please check them.
4. In P10, the experimental results shown in Fig. 4 a-b are not clear. Please enhance the contrast of the image, as well as Fig. S5 and S6 in the supplementary information.
5. In P10, "each moments" should be "each moment".
6. In P14, the green scale bar in Figure 7 is not clear.
7. In P16, the scale bar of Fig. 9 (a) is not visible. The reviewer suggests removing it to the top layer.
8. In SP5, samples with loading time should be changed into samples versus loading time.
9. In SP15, "no Luders bands are observed in the experiments" the reason is absent, please supplement.

Reviewer #3 (Remarks to the Author):

The revised manuscript is indeed improved and the authors have partially addressed my concerns. Additional experimental data were also added to support their conclusions and the damage model. There are new discoveries on the strain induced magnetic flux penetration presented in this paper.

However, compared to the previous reports, their improvements on the experimental device are not significant, and I do not think the mechanism of the flux penetration is very different. It is well known that magnetic flux will penetrate along the defects where the flux pinning force is weakened. In Ref.22, the MOI images (Fig.6 and 7 there) showed the magnetic flux penetrations along the grain

boundary or defects, and in this manuscript, the magnetic flux penetrations are also related with the strain induced internal sample damage although the damage may start from the substrate and be extended to the sample. In this sense, the mechanism of flux peneataion is not different in the two studies. Moreover, the flux penetration patterns are used to track the internal sample damage in both studies.

Responses to referees

We are very grateful to the reviewers for taking the time and effort to review our paper. Their positive comments and insightful suggestions have helped us to improve the manuscript. According to their comments and suggestions, we have revised this article and corrected the spelling, grammar, word use, and punctuation throughout our manuscript. The point-by-point responses to each critique are provided below.

To reviewer 2

Query 1: For the first query of the reviewer, “In MO image results, the center part is clear but color turn into darkening at the corner, but the width of mixed-state seems the same along the length of the CC, please give the explanation.” The explanation given by the authors is that it is caused by the non-uniform illumination, I approve this response, but I still want to know how to eliminate it? Figure 3 displayed the results of local magnetic field variation. Do the authors consider eliminating the influences of non-uniform illumination on the present results? Please clear it.

Answer: The results of local magnetic field variation (Fig. 3c) have eliminated the influences of non-uniform illumination. The calibration method is conducted as follows:

The intensity in the MO image plane can be written as

$$I(x, y) = I_1(x, y) \sin^2(\alpha) + I_0(x, y), \quad (1)$$

where the α is the rotation of the polarization plane as a function of magnetic field B . Non-uniform illumination exists as the terms of the amplitude of $I_1(x, y)$ and back ground of $I_0(x, y)$, since these two terms are both independent of magnetic field, a fast calibration method can be built as followed: firstly, MO images of tapes are recorded by the camera; After that, elevate the temperature over the T_c so that the flux can freely penetrate the sample; Next, multiple images are taken for calibration by step wisely increasing the magnetic field, in this step, because the whole field of view covers just about $3.6 \times 2.7 \text{ mm}^2$ considering the low deviation of magnetic field in this little area, the applied magnetic field can be treated as a uniform, therefore the uneven intensity in these images is just caused by the non-uniform illumination. In each calibration image, the intensity $I_c(x, y)$ can be expressed as

$$I_c(x, y) = I_1(x, y) f_1(B_a) + I_0(x, y), \quad (2)$$

Where $f_1(B_a)$ a field dependence of function, the distribution $I_0(x, y)$ of can be obtained without applied field, as well as the max intensity $I_{c_{\max}}(x, y)$ with max applied magnetic field can be written as

$$I_{c_{\max}}(x, y) = I_1(x, y) f_1(B_{c_{\max}}) + I_0(x, y). \quad (3)$$

Using the image taken under the max applied magnetic field as a reference image, one could get a unique normalized function just dependent on magnetic field

$$\left[100 \frac{I_c(x, y) - I_0(x, y)}{I_{\text{cmax}}(x, y) - I_0(x, y)} \right] = \frac{f_1(B_a)}{f_1(B_{a\text{max}})} = f_2(B_a). \quad (4)$$

This means that the calibration function $f_2(B_a)$ can be obtained through a series of calibration images, inversely, $f_2^{-1}(B_a)$ can be plotted; using a fitting of a polynomial of $f_2^{-1}(B_a)$, one can extract the magnetic field from intensity of the image

$$B_z = f_2^{-1} \left[100 \frac{I_c(x, y) - I_b(x, y)}{I_{\text{cmax}}(x, y) - I_b(x, y)} \right]. \quad (5)$$

Using the MO image of superconducting tape with intensity of $I(x, y)$, one can pixel-wisely reconstruct the magnetic field distribution as

$$B_z(x, y) = f_2^{-1} \left[100 \frac{I(x, y) - I_0(x, y)}{I_{\text{cmax}}(x, y) - I_0(x, y)} \right] \quad (6)$$

This method has been verified by experimental results, and details can be found in our published work (see Ref. 26 in the revised manuscript).

Query 2: In P18, the discussion of the relationship between “spot-like flux motion” and the “collective flux motion” should be deleted. Since the present flux motion caused by strain is a new experimental phenomenon, the reviewer considers that the experimental observations of the spot-like flux motion in this paper are significantly different from the classical flux creep considered in the condensed matter physics. If the authors want to retain this discussion, please explain in depth.

Answer: We have accepted your suggestion. The discussion of the relationship between “spot-like flux motion” and the “collective flux

motion” has been deleted in the revision.

Query 3: There is still some syntax and semantics in this revised version, In P5, application should be applications, move should be moves, etc. Please check them.

Answer: We have fixed the text problems mentioned above, and checked the remaining text to make sure the language expression is accurate.

Query 4: In P10, the experimental results shown in Fig. 4 a-b are not clear. Please enhance the contrast of the image, as well as Fig. S5 and S6 in the supplementary information.

Answer: The experimental results shown in Fig. 4 (a)-(b), and those in Fig. S5 and S6 (the supplementary information) are all cropped from the *raw experimental magneto-optical images*. The reason that these cropped parts compared with the inserted in Figs. 3a and 3b have lower resolution and contrast is that the images of high speed camera of NR4-S1 has 1024×1024 pixels per image (with resolution of $4.5\mu\text{m}/\text{pixel}$) and time of exposure is 5ms, while the color camera has 1288×972 pixels per image (with resolution of $2.8\mu\text{m}/\text{pixel}$) and time of 100ms. To show and keep the as-observed characteristic of the local flux motion, we decide not to handle these raw images artificially but to resize and crop raw images (we also handle others images with the same rule). However, to give a better

visualization we take a difference between the local-flux-motion image and the reference image, as shown Figs. (e)-(g). The similar process is also conducted in Figs. S5 and S6.

Query 5: In P10, “each moments” should be “each moment”.

Answer: It has been corrected.

Query 6: In P14, the green scale bar in Figure 7 is not clear.

Answer: The scale bar has been replaced with a clear one.

Query 7: In P16, the scale bar of Fig. 9 (a) is not visible. The reviewer suggests removing it to the top layer.

Answer: The scale bar of Fig. 9 (a) has been removed to the top layer.

Query 8: In SP5, samples with loading time should be changed into samples versus loading time.

Answer: It has been corrected.

Query 9: In SP15, “no Luders bands are observed in the experiments” the reason is absent, please supplement.

Answer: The reason has been added as “A tensile experiment for pure Hastelloy-C276 substrate with 4mm width is conducted at 40K with

white light illumination, one can find no strip-like patterns that extended along tape width direction with nearly 45 degrees with respect to the tape length, hence no Luders band is observed at the surface as displayed in Fig. S18. The main reason is that the observation area of the magneto-optical is far away from the chuck, and the strain of the material is uniform before the crack appears, so it does not appear the Luders band.”

To reviewer 3

The revised manuscript is indeed improved and the authors have partially addressed my concerns. Additional experimental data were also added to support their conclusions and the damage model. There are new discoveries on the strain induced magnetic flux penetration presented in this paper.

Query 1: However, their improvements on the experimental device are not significant, and I do not think the mechanism of the flux penetration is very different. It is well known that magnetic flux will penetrate along the defects where the flux pinning force is weakened. In Ref.22, the MOI images (Fig.6 and 7 there) showed the magnetic flux penetrations along the grain boundary or defects, and in this manuscript, the magnetic flux penetrations are also related with the strain induced internal sample damage although the damage may start from the substrate and be

extended to the sample. In this sense, the mechanism of flux penetration is not different in the two studies. Moreover, the flux penetration patterns are used to track the internal sample damage in both studies.

Answer: The authors are very grateful to the reviewer for the approval of our supplementary experiment and revised manuscript. Now, according to the review's concerns and above-mentioned insightful comments, we further emphasize the meaningful merits of our research as following:

We indeed acknowledge the pioneering work done for micro-superconducting bridge on the response of magnetic flux during mechanical loading by the MOI method in Ref. 22, as pointed in our introduction part. However, as we know, for practical application, e.g., cables and magnet coils, CCs must be applied with engineering scale of length in meters and width in millimeters. Therefore, experiment result for samples with micro-superconducting bridge is not enough to fully reveal the strain-induced damage in a macro-scale CC during loading, which motivates and enlightens us to build a specific instrument for a macro-scale CC testing. Though the functions of device that we constructed and that built in Ref. 22 have similarity, but the instrument used in this work have several totally differences compared with that in Ref. 22, as below,

1. We completed the design, construction and validation of cryogenic-in-field loading system for macro-scale CC samples with

long length and wide width. For doing this, we have solved the challenges of cooling system based on GM refrigerator (cooling power become lower when temperature gets lower) by special thermal conduction and insulation design, and the challenges of loading system with large loading range but with small space occupation.

2. We have built a special objective focusing module for incorporating the magneto-optical imaging into the loading system, which eliminates the reduction of the resolution caused by refraction from the glass window, and the space limit between the sample and the optical microscope.
3. In the aspect of mechanism of the flux penetration, though both two works are dedicate to the mechanism revealing on flux penetration caused by tensile strain. However, flux penetration is attributed to damage-driven by the Luders bands in Ref. 22. The Luders-bands induced crack were just found near the clamp and no cracks were observed away from the clamp, as pointed out by the writers and shown in Fig. 5 of Ref. 33. Thus damage-driven flux penetration by the Luders bands found in Ref. 22 may be caused by a non-homogenous strain distribution within the sample with the limited sample size. We think it is similar to the case that non-uniform strain forms when closed to the clamp and this can explain the lower fracture strain value (about 0.15%) reported in Ref. 22 and that of macro-scale

experiment (0.3%) reported in Ref. 33.

4. Moreover, the magnetic field dependence of strain-induced flux penetration is not investigated in Ref. 22. As a contrast, with the new instrument used, the flux penetration in our work is attributed to the crack that extended from the tensile substrate and the strong field dependence of strain-induced flux penetration is firstly found to the best of our knowledge.

At last, though both work used the same methodology that using flux penetration patterns to track the internal sample damage, nevertheless, from the viewpoint of novelty creation, the work in Ref. 22 emphasizes a validation of the damage of Luders bands, we have some new discoveries including thickness dependent crack induced by strain and amorphous phase near the crack tip.

REVIEWERS' COMMENTS

Reviewer #2 (Remarks to the Author):

This revised manuscript is much improved and the responses to what I concerned about are adequate. For this revised version, I have no additional comments to offer, and in my opinion, it should reach the requirements for publication.